# Microbial diversity of coastal microbial mats formations in karstic habitats from the Yucatan Peninsula, Mexico

Santiago Cadena[1], Claudia Teutli-Hernández[2], Jorge A. Herrera-Silveira[3], M. Leopoldina Aguirre-Macedo[3], Luisa I. Falcón[4], Brad M. Bebout[5], José Q. García-Maldonado [3]*

1 Centro de Investigaciones Químicas, Universidad Autónoma del Estado de Morelos, Cuernavaca, Morelos, Mexico, 2 Escuela Nacional de Estudios Superiores Mérida, Universidad Nacional Autónoma de México, Ucú, Yucatan, Mexico, 3 Centro de Investigación y de Estudios Avanzados del Instituto Politécnico Nacional, Departamento de Recursos del Mar, Unidad Mérida, Yucatan, Mexico, 4 Departamento de Ecología Evolutiva, Instituto de Ecología, Unidad Mérida, Universidad Nacional Autónoma de México, Ucú, Mexico, 5 Exobiology Branch, Ames Research Center, National Aeronautics and Space Administration, Moffett Field, California, United States of America

* jose.garcia@cinvestav.mx

**Data availability statement:** The 16S raw sequences generated in this study were deposited in the NCBI Sequence Read Archive under the BioProject PRJNA971606.

**Funding:** This study was funded by Consejo Nacional de Ciencias, Humanidades y

## Abstract

In this study, we report for the first time an exploration of the physicochemical characteristics and the prokaryotic diversity of three different types of microbial mats from karstic habitats located in Sisal, Progreso and Ría Lagartos, in the Yucatan Peninsula, Mexico. Our results showed that lift-off mats were found in the lower salinity (2.2%) area (Sisal), while flat and pustular mats were detected in hypersaline (6–9%) sites (Progreso and Ría Lagartos). Notably, some of these microbial mat structures were in close proximity to mangrove forest ecosystems with both degraded and restored regimes. XRD analysis revealed different mineral compositions of the mats; however, aragonite, calcite, and halite were commonly found in all samples studied. High-throughput sequencing of the 16S rRNA gene identified differences in microbial communities across the different mat types, and statistical analyses revealed that salinity, redox potential, and temperature were significant factors in explaining the variance of the prokaryotic assemblages. Microbial groups identified in this study include those known to be important in the biogeochemical cycling of key elements, such as carbon, nitrogen, and sulfur. Interestingly, the community composition of flat and pustular mats from Progreso was similar, with Bacteroidia, Anaerolineae, and Phycisphaerae being the most abundant microbial groups in flat mats; and Bacteroidia, Anaerolineae, and Alphaproteobacteria dominating pustular mats. By contrast, flat mats from Ría Lagartos were dominated by Halobacteria, Cyanobacteria and Bacteroidota, while Bacteroidia, Gammaproteobacteria and Cyanobacteria dominated lift-off mats from Sisal. This work contributes to understanding the distribution, physicochemical characteristics and microbial diversity of coastal microbial mats,

Tecnologías (CONAHCYT) project CF-2019-848287 awarded to JQGM. https://conahcyt.mx/ . In addition, sampling was performed thanks to Programa de Apoyo a Proyectos de Investigación e Innovación Tecnológica (PAPIIT-UNAM) IN216219 awarded to CTH. https://dgapa.unam.mx/ the funders had no role in study design, data collection and analysis, decision to publish, or preparation of the manuscript.

**Competing interests:** The authors have declared that no competing interests exist.

providing valuable new insights into microbial mats that develop in karstic ecosystems. This information is relevant to ongoing and future efforts to manage and preserve coastal ecosystems in the Yucatan Peninsula.

## Introduction

Microbial mats are intricate associations of multiple functional groups of microbes that grow on surfaces such as sediments, rocks and sand [1]; they can be found in diverse aquatic environments including hot springs, hypersaline ponds, alkaline lakes, dry and hot deserts, and coastal intertidal areas [2]. Mats typically flourish in zones with minimal competition from plants and grazing organisms [3], possibly due to the adverse effects of nutrient-poor conditions, and only periodic inundation, leading to desiccation and significant fluctuations in temperature and salinity [1–3].

Microbial mats can be differentiated based on their complex morphologies. These morphologies include the typical flat mat, but also include pustular, gelatinous, tufted, reticulate, and blister mats, among others [4–6]. Mat formation is shaped by diverse abiotic factors, such as water availability, sediment influx, wind intensity and type of substrate [3,4,7]. In addition, it is known that different mat morphologies contain diverse microbial communities [8–9]. Up to now, the most examined types of coastal mats are located in Australia. Flat microbial mats from Shark Bay have been reported to be dominated by archaeal communities from Parvarchaeota and Thermoplasmata, while pustular mats are dominated by Halobacteria [10]. Pustular mats from Hamelin Pool are dominated by the Cyanobacteria *Entophysalis*, compared to flat mats that usually contain *Coleofasciculus* [11]. Proteobacteria, Bacteroidetes and Planctomycetes are the most common taxa in pustular mats from Carbla Beach, where Cyanobacteria from Pseudophormidiales and Phormidesmiales are responsible for the production of sulfated extracellular polymeric substances [9]. D'Agostino and colleagues [12] investigated flat, pustular and tufted mats from Hamelin Pool indicating that the mat's shape helps to protect microbial communities from ultraviolet radiation and environmental pressures. Eukaryotic diversity in these two mat types is also different; flat mats can contain nematodes, fungi and diatoms while pustular mats harbor fungi, diatoms, tardigrades and microalgae [8].

While some studies have investigated microbial diversity and function in hypersaline ecosystems, microbial mats in karstic environments have received comparatively less attention. Kolda and colleagues [13] investigated microbial mats from the Krčić spring, Croatia, finding that Cyanobacteria constituted the core structure of the community, reporting *Microcoleus* and *Phormidium* as the most abundant cyanobacterial taxa, followed by several groups of Alphaproteobacteria and Firmicutes. Mulec and Engel [14] studied the microbial diversity of mats along an oxygen-sulphide gradient in the Žveplenica karst spring in Slovenia, finding abundant sulphur-metabolizing taxa such as *Thiothrix*, *Azospira*, *Iodobacter* and *Thiobacillus*. Microbial mats from karst springs in Lower Kane Cave, Wyoming, were investigated using 16S rRNA gene clone libraries, reporting chemolithoautotrophic

taxa related to Epsilon, Gamma and Beta Proteobacteria, as well as *Acidobacterium* and *Chlorobi* [15]. Compared to karstic microbial mats studied in caves and springs, coastal mats are exposed to different stressors, such as tidal fluctuations, salt intrusion, and mangrove-derived organic matter [1–3], which provide a distinct ecological niche for microbial mats.

The Yucatan Peninsula (YP) is one of the most extensive karst systems known on the planet, encompassing a limestone platform of 165,000 km$^2$, with a network of cenotes, caves, and coastal lagoons [16]. The climatic conditions of the YP are divided into three seasons: dry (March-June), rainy (July-October), and north-winds (November-February). Coastal karstic habitats in the YP are influenced by dynamic environmental factors, such as salinity gradients, seasonal precipitation, and the proximity of mangrove ecosystems [17]. The hypersaline conditions resulting from high evaporation rates in these coastal zones often lead to the formation of solar salterns, which may harbor microbial mats similar to those in other hypersaline environments [18]. However, the specific microbial communities inhabiting these karstic mats remain poorly characterized. This study explores the prokaryotic diversity of microbial mats in three coastal karstic habitats in the Yucatan Peninsula: Sisal, Progreso, and Ría Lagartos. Using high-throughput sequencing of the 16S rRNA gene, we characterize the microbial community composition and identify key environmental drivers, including salinity, temperature, and redox potential. By comparing these karstic mats with other coastal and hypersaline ecosystems, this research aims to provide new insights into microbial mats' ecological and functional roles in understudied karstic coastal habitats, contributing to broader efforts to conserve and manage these unique ecosystems.

## Materials and methods

### Sampling and physicochemical characterization

Microbial mats were collected in May 2019 from three coastal zones in Yucatan, Mexico, specifically from Sisal, Progreso, and Ría Lagartos (Fig 1). The source for this map is the 2024 Digital Topographic Map of Mexico by INEGI (https://gaia.inegi.org.mx/mdm6). Notably, some microbial mats were found close to degraded mangrove forests, and some are being managed and restored from degraded conditions. Degraded mangroves refer to areas that previously supported mangrove vegetation but had widespread loss, resulting in the absence of mangrove cover. Restored mangroves are sites where previously degraded areas have undergone ecological rehabilitation, characterized by the reestablishment of mangrove seedlings through active restoration efforts [19]. At each locality, triplicate surficial mat cores measuring 8 cm in width by 8 cm in length and 3–5 cm in depth were collected. Subsequently, from each of the previously obtained cores, three sub-cores from the upper first centimeter of the mats (1 cm × 1 cm) were taken to obtain 9 representative samples per locality. The sub-cores were immediately stored in liquid nitrogen for sequencing. At each sampling site, interstitial water was extracted using a specialized device consisting of a 50 ml syringe connected to an acrylic tube. The tube, sealed at the bottom except for a small hole (3 mm diameter), was inserted to a depth of 30 cm within the mats. Water was drawn out by syringe suction, yielding 40 ml per sample while minimizing exposure to air [20]. Immediately following collection, physicochemical properties, including salinity, temperature, pH, and redox potential, were recorded using a portable multi-parameter instrument. Each environmental variable was measured 9 times at each site, except for Ría Lagartos, where technical difficulties on site only allowed for a single data point to be collected for each variable.

### Mineralogical analysis by XRD

Samples of microbial mats were dried at room temperature, then finely macerated (<20 um) and placed into a muffle furnace for 12 hours at 550°C to calcinate organic materials. To determine the mineral composition of the microbial mats, samples were subjected to X-ray diffraction (XRD) analysis using a Bruker D-8 Advance (IUC, Indore) diffractometer with a Cu tube (kα: 1.5406 Å) operated at 30 mÅ and 40 kV, at CINVESTAV-Mérida.

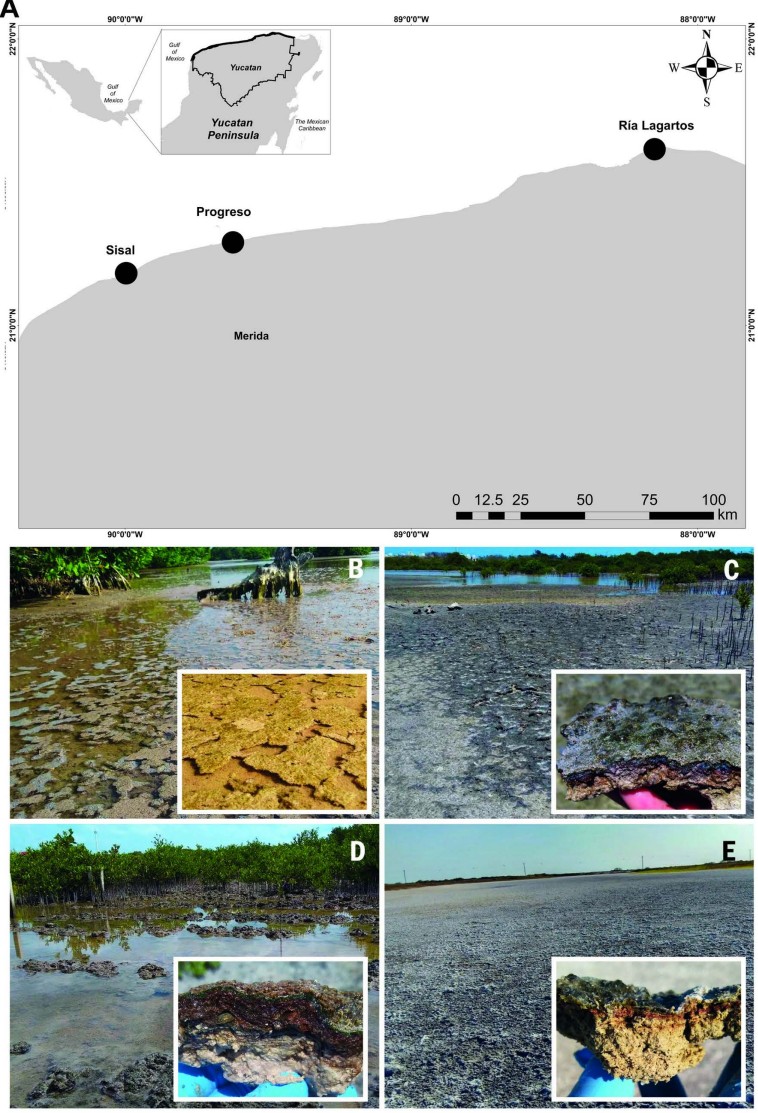

**Fig 1. Sampling sites of microbial mats from the Yucatan Peninsula.** Map showing sampling sites **(A)**. Source: INEGI, Digital Topographic Map, 2024, México (https://gaia.inegi.org.mx/mdm6). The original maps were modified to show the study sites. Map modifications comply with terms of use required by INEGI for free distribution (https://www.inegi.org.mx/contenidos/inegi/doc/terminos_info.pdf). Different types of coastal microbial mats were documented: Lift-off mats from Sisal (B), Pustular (C) and flat (D) microbial mats from Progreso, and flat mats from Ría Lagartos (E).

### DNA extraction, amplicon library construction for sequencing and bioinformatics analysis

DNA of collected microbial mats from each locality (n = 9) was extracted from 0.25 g of microbial mat sample using the PowerSoil DNA Isolation Kit (Mo Bio Laboratories, Carlsbad, CA, USA) following the conventional instructions, with cell lysis carried out using a TissueLyser LT (Qiagen, Hilden, Germany). A blank (a column with no sample supplied) was processed alongside the extractions to monitor potential contamination. DNA quality was assessed by running samples on a 1% agarose gel.

To amplify 16S rRNA gene fragments from Bacteria and Archaea, universal primer sets 515F-Y and 926R were used, which cover the V4-V6 regions [21]. The primer characteristics and thermocycling conditions were detailed by Needham

and colleagues [22]. PCR reactions (20 µl) were performed using 2 µl of DNA (5 ng/µl), 0.5 µl of each primer (10 µM), and 10 µl of 2 × Phusion High-Fidelity MasterMix (Thermo Scientific, Waltham, MA, USA). PCR fragments were first purified using magnetic beads (AMPure XP) from Beckman Coulter Genomics (Brea, CA, USA). Subsequently, amplicons were indexed using the Nextera XT Index kit version 2 (Illumina, San Diego, CA, USA), following the Illumina 16S Metagenomic Sequencing Library Preparation protocol. After indexing, barcoded fragments were purified a second time and quantified using the Qubit 3.0 fluorometer from Life Technologies (Malaysia). The correct size of PCR products was confirmed on an Advanced QIAxcel from QIAGEN (USA). Barcoded PCR-amplicons were diluted in 10 mM Tris (pH 8.5) and pooled in equimolar concentrations (9 pM). Finally, paired-end sequencing was performed on the MiSeq platform (Illumina, San Diego, CA, USA) with a MiSeq Reagent Kit V3 (600 cycles), using the Aquatic Pathology Laboratory at CINVESTAV-Mérida. All sequence data is publicly available in the NCBI Sequence Read Archive under the BioProject PRJNA971606.

The QIIME2 (2017.11) pipeline [23] was used to analyze demultiplexed reads. DADA2 plugin was employed to denoise and resolve amplicon sequence variants (ASV), and chimeras were eliminated using the "consensus" method [24]. Representative ASVs were assigned using the SILVA small subunit ribosomal RNAs (16S) v138 database as a reference, since SILVA provides one of the most updated and extensive collections of 16S gene sequences derived from publicly available datasets [25], through the V-SEARCH plugin [26]. Representative reads were aligned with the MAFFT algorithm [27] and filtered using fasttree [28] to build a phylogenetic tree. The data were normalized by sub-sampling to the lowest read count per sample (10,700). R environment was used for graphic visualization and statistical analysis, with phyloseq [29] and ggplot2 [30] libraries. Additionally, alpha diversity (observed ASVs, Shannon and Simpson indexes) and beta diversity from samples were calculated. One-way ANOVA was performed to determine the significance of the differences between alpha diversity measurements, followed by a post hoc Tukey's test when significant differences were identified. Furthermore, in order to characterize the most representative ASVs of each type of microbial mat, we used a Linear Discriminant Analysis (LDA) Effect Size (LEfSe) [31], employing an LDA threshold >2, coupled with internal Kruskal-Wallis and Wilcoxon tests, ensuring significance at a p-value <0.05, performed in the Galaxy web application [32]. The PICRUSt2 v.2.2.0 [33] pipeline was used to infer metagenomic functional content based on QIIME2 microbial community profiles obtained from representative sequences and sequence features. Predicted functional genes were categorized using the KEGG ontology database (http://www.genome.jp/kegg/). STAMP software (v2.1.3) was used for statistical analyses comparing community structures between localities at different levels (Welch's t-test); and LEfSe analysis was used to assess differential functional abundance based on predicted KEGG pathways (LDA score 4.0 and p-value < 0.05).

This field research was permitted by the Subsecretaría de Política Ambiental y Recursos Naturales, Dirección General de la Vida Silvestre under the permission SPARN/DGVS/04857/23 which allowed access to protected areas. Protected species were not sampled for this research.

## Results

### Physicochemical characteristics

Different types of microbial mats were found and denominated as lift-off mats, pustular, and flat mats. Salinity measurements at the study sites ranged from 2.2% to 9.8%, with the lowest values found at Sisal and the highest at Ría Lagartos. The average temperature of the interstitial water varied from 31 to 37°C. pH values were similar across all sites, with an average of 7.3 ± 0.2. The redox potential measurements ranged from −177 to −299 mV. The physicochemical properties of the microbial mats that were sampled are provided in Table 1.

Nine different minerals were detected by XRD analysis of microbial mats (Fig 2). Calcite and halite were present in all the studied microbial mats, including the lift-off mats, flat, and pustular mats. In addition, beidellite and quartz were only found in the flat mats from Progreso, highlighting the differences in mineral composition across the different mat types. A more detailed comparison of the mineral composition of the microbial mats is shown in S1 Table.

**Table 1. Physicochemical characterization.**

| Locality | Macrostructure | Type of associated mangrove | Salinity (%) | Temperature (°C) | pH | Redox potential (mV) |
|---|---|---|---|---|---|---|
| **Sisal** | Lift-off mats | Degraded | 2.2 (±0.22) | 31.4 (±0.72) | 7.4 (±0.07) | −299.6 (±20.08) |
| **Progreso** | Pustular mat | Restored | 6.0 (±0.36) | 34.6 (±1.29) | 7.2 (±0.06) | −271.1 (±21.07) |
| **Progreso** | Flat mat | Degraded | 8.8 (±0.54) | 32.4 (±1.63) | 7.4 (±0.11) | −177.7 (±60.33) |
| **Ría Lagartos** | Flat mat | No mangrove | 9.8 | 37.3 | 7.2 | −207.0 |

Sampled sites and type of macrostructure.

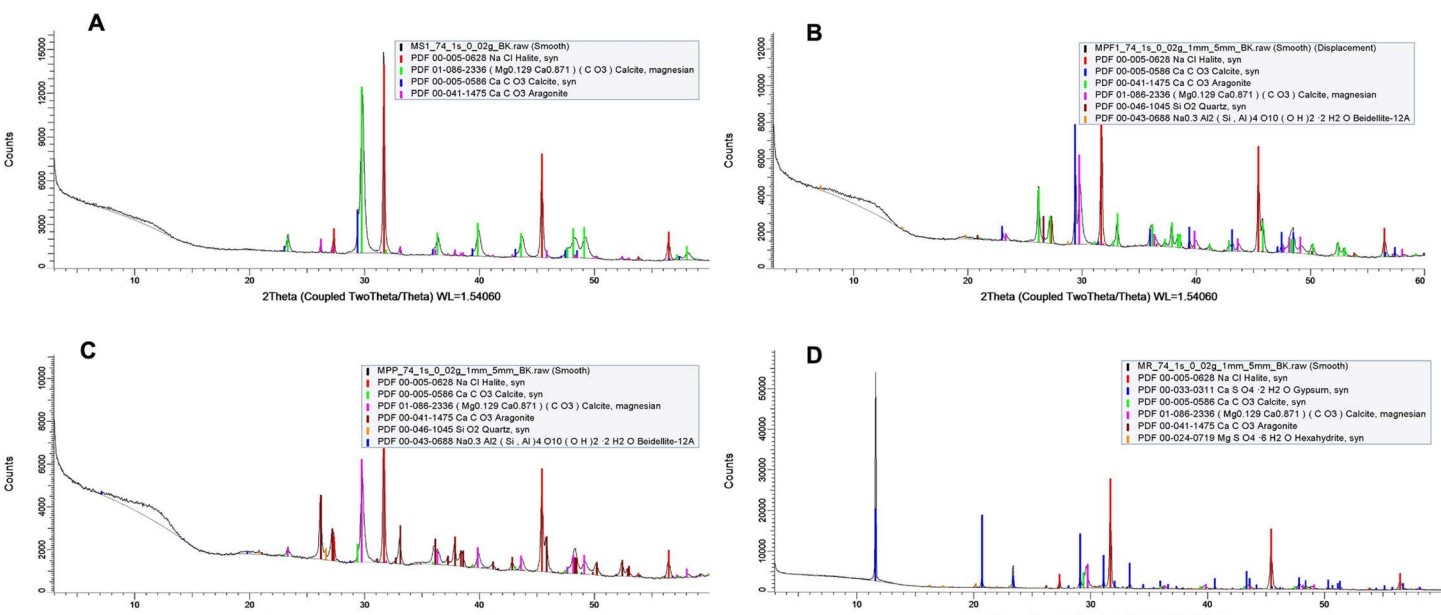

**Fig 2. X.ray diffraction patterns of microbial mat samples collected from different locations.** Lift-off mats from Sisal (A); flat (B) and pustular (C) mats from Progreso and flat mats from Ría Lagartos (D).

## Microbial community structure analysis

Samples of all microbial mats were sequenced including lift-off mats from Sisal, pustular and flat mats from Progreso, and flat mats from Ría Lagartos. Sequencing of 16S rRNA gene obtained 870,175 raw reads from studied mats, yielding a total of 751,471 high quality sequences. Quality filtering is shown in S2 Table. The observed amplicon sequence variants (ASVs) ranged from 729 to 1081, and the Shannon diversity index ranged from 5.76 to 6.23, while the Simpson index varied between 0.992 and 0.996 (Fig 3A). ANOVA showed significant differences in alpha diversity measurements (observed ASVs, Shannon and Simpson indexes) among the sampling locations ($p < 0.001$). Tukey's test further revealed that Ría Lagartos had significantly lower diversity compared to the other sites ($p < 0.001$). Principal Coordinates Analysis (PCoA) based on the weighted and unweighted UniFrac distances demonstrated that the microbial community structure of the mats was distinct among the different sampled sites (Fig 3B and 3C).

The results of the PERMANOVA analysis between environmental data and relative abundance of ASVs indicated that salinity, temperature, and redox potential influenced microbial community structure of microbial mats studied (S3 Table). Specifically, salinity was found to explain 17% of the variance, while temperature and redox potential explained 13% and

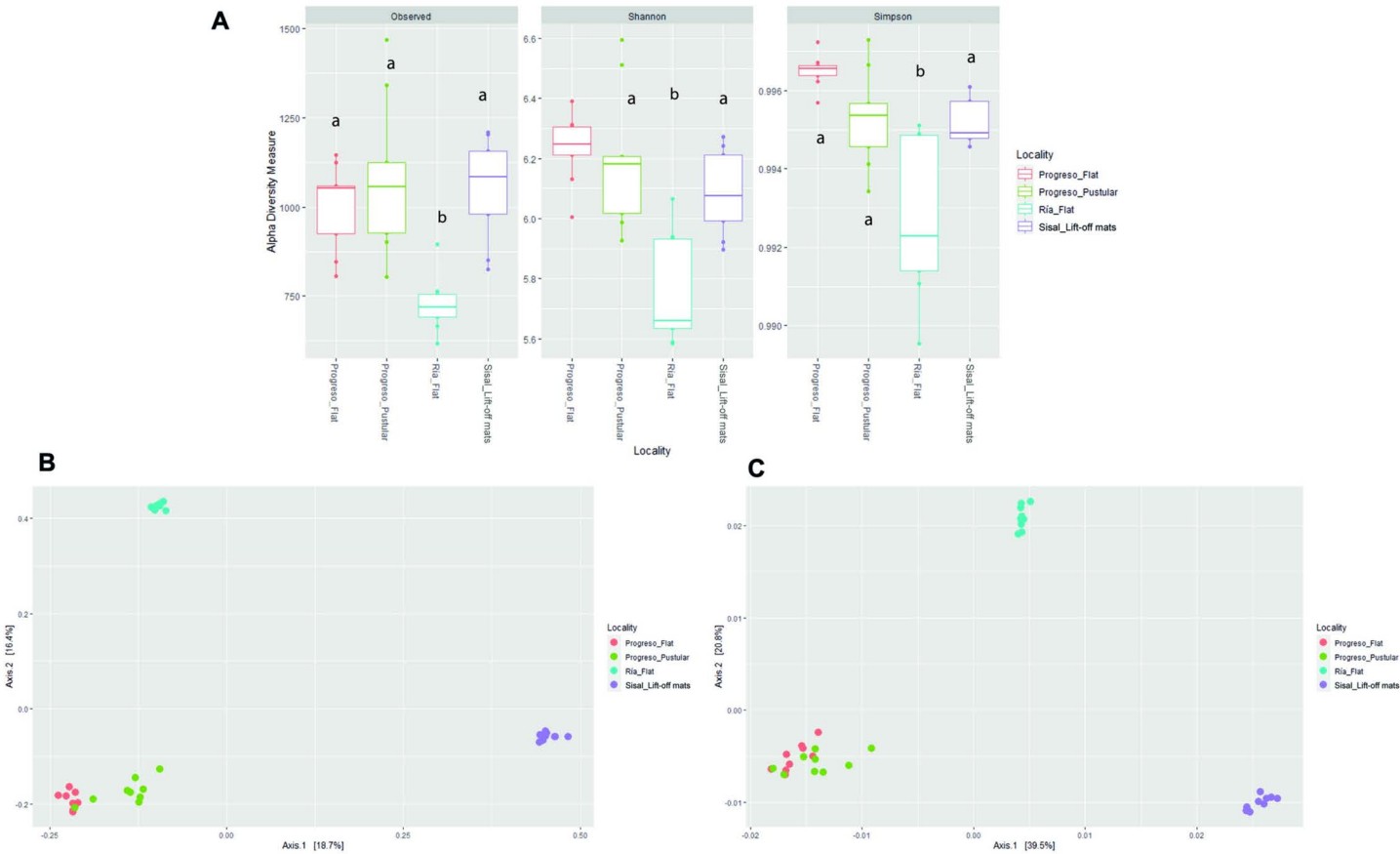

**Fig 3. Alpha and beta diversity of microbial mats samples.** Observed ASVs, Shannon and Simpson indexes (A). Localities that are significantly different are denoted with different letters (a, b or c) (one-way ANOVA, Tukey's test, $p < 0.001$). PCoA calculated on the unweighted (B) and weighted (C) UniFrac metrics, based on 16S rRNA gene amplicon sequences.

10%, respectively, indicating the importance of these environmental variables in shaping the composition of microbial communities in these mats.

## Microbial community composition and predicted pathways

The archaeal communities detected in microbial mat samples were affiliated with nine phyla, including Halobacterota, Nanoarchaeota, Asgardarchaeota, and Thermoplasmatota as the most represented. Archaeal communities in microbial mats from Sisal and Progreso ranged from 3% to 7% of the total ASVs. In contrast, mats from Ría Lagartos contained between 20% to 35% of Halobacterota.

The bacterial community in the microbial mats was highly rich, covering 39 phyla, but was dominated by Bacteroidetes, Proteobacteria, Chloroflexi, Planctomycetota, and Cyanobacteria ([Fig 4]). Bacteroidetes (40−13%) was composed of several classes, including Bacteroidia (37−5%), Rhodothermia (17−1%), and Ignavibacteria (2–0.2%). Proteobacteria (23−7%) contained members from Gammaproteobacteria (19−3%) and Alphaproteobacteria (3−6%) classes, while Chloroflexi (28−4%) included Anaerolineae (21−3%) and Chloroflexia (7−1%). Planctomycetota (14−2%) was mainly represented by Phycisphaerae (9−1%). Finally, Cyanobacteria accounted for (19−5%) of diversity.

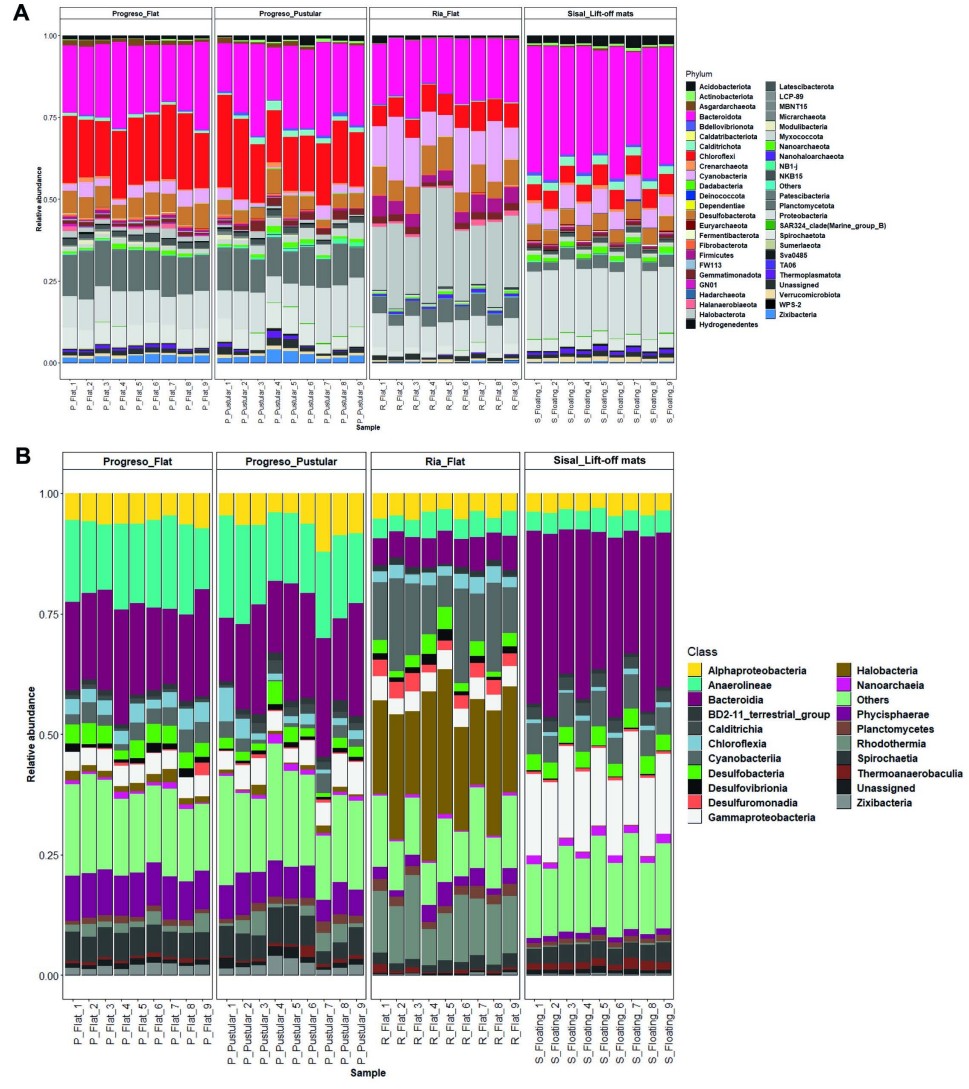

**Fig 4. Relative abundance of taxonomic groups of microbes observed.** Stacked bars at phylum (A) and class (B) level from studied microbial mats. Clades represented <1% were grouped in "Others".

The PCoA analysis indicated that the community composition of flat and pustular mats from Progreso were similar to each other (Fig 3B and C). In flat mats from Progreso, Bacteroidetes (Bacteroidia, 22−17%), Chloroflexi (Anaerolineae, 19−12%), and Planctomycetes (Phycisphaerae, 8−9%) were the most abundant microbial groups. In comparison, pustular mats from the same location were dominated by Bacteroidetes (Bacteroidia, 2−13%), Chloroflexi (Anaerolineae, 21−14%), and Proteobacteria (Alphaproteobacteria, 4−12%). In contrast, flat mats from Ría Lagartos were dominated by Halobacterota (Halobacteria, 35−17%), Cyanobacteria (19−6%), and Bacteroidota (Rhodothermia, 17−7%). Lift-off mats from Sisal were found to contain Bacteroidetes (Bacteroidia, 37−25%), Proteobacteria (Gammaproteobacteria, 16−19%), and Cyanobacteria (7−5%) as the most abundant microbial groups (Fig 4). Detailed information in the microbial richness detected on mats at the ASV level is presented in S4 Table.

LefSe analysis identified the most representative ASVs from each studied microbial mat. Candidatus Chlorothrix, Phycisphaeraceae, AKAU3564 sediment group, A4b, and Sb_5 were the most representative taxa in flat mats from

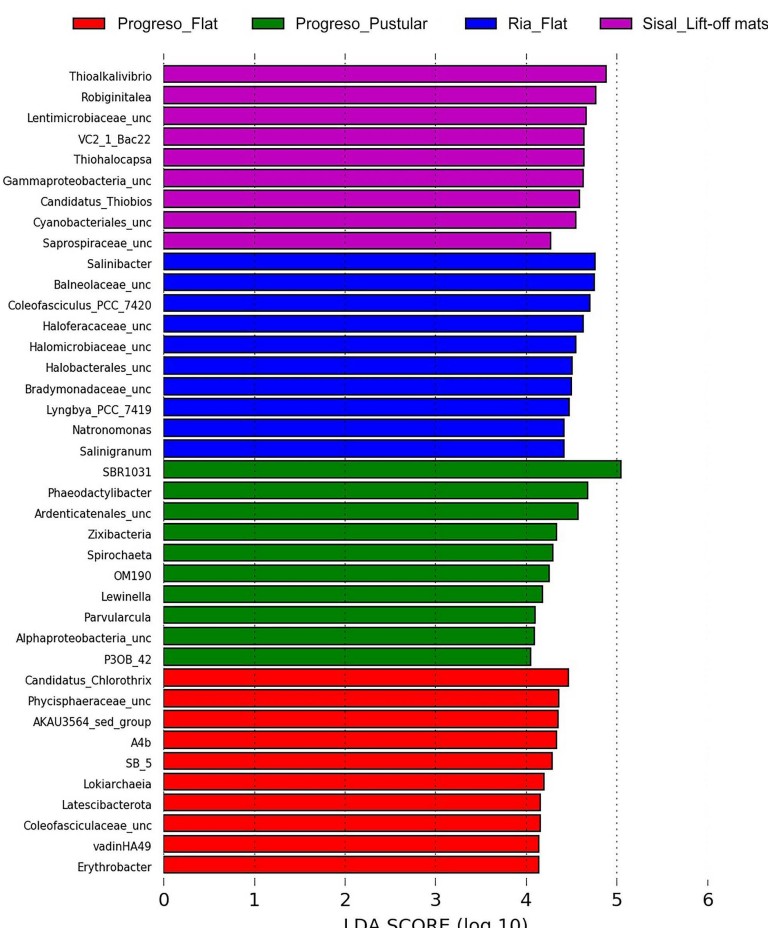

Progreso, while SBR1031, Phaeodactylibacter, Ardenticatenales, Zixivacteria, and Spirochaet characterized pustular mats from Progreso. Flat mats from Ría Lagartos displayed *Salinibacter*, *Balneolaceae*, *Coleofasciculus_PCC_7420*, Haloferacaceae, and Halomicrobiaceae, and lift-off mats from Sisal had *Thioalkalovibrio*, *Robiginitalea*, *Lentimicrobiaceae*, VC2_1 Bac22, and *Thiohalocapsa* as the most represented taxa. Fig 5 shows the top 10 taxa with the highest LDA scores.

Functional prediction of microbial communities derived from PICRUSt2 analysis allowed to identify 409 metabolic pathways among all microbial mat samples (S5 Table). Fig 6 shows the predicted metabolic pathways for each microbial mat type, revealed by LefSe analysis. Flat mats from Progreso showed higher proportions of nucleosides, nucleotides and carbohydrates biosynthesis, while Pustular mats from Progreso showed polyprenyl biosynthesis and pentose phosphate pathways. Lift-off mats from Sisal possessed differential abundances of fatty acids, lipids and cell structure biosynthesis. Flat mats from Ría Lagartos were characterized by vitamins, cofactors and, amino acid biosynthesis, inorganic nutrient metabolism and carbohydrate degradation. In addition, amino acid metabolism was significantly higher in lift-off mats from Sisal. Methane metabolism was statistically significant in flat mats from Ría Lagartos, and sulfur and carbon fixation metabolisms were not statistically different among microbial mat types (S1 Fig).

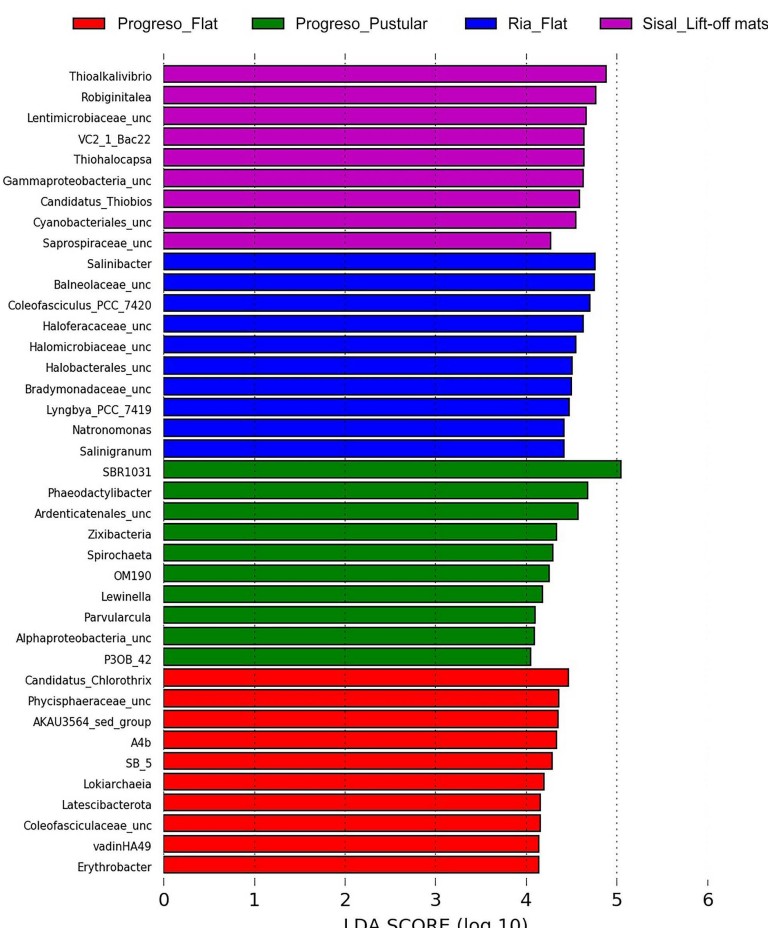

**Fig 5. LefSe analysis based on retrieved 16S rRNA sequences from microbial mat samples.** The top 10 organisms with the highest LDA scores are shown, indicating their potential as biomarkers for each mat type.

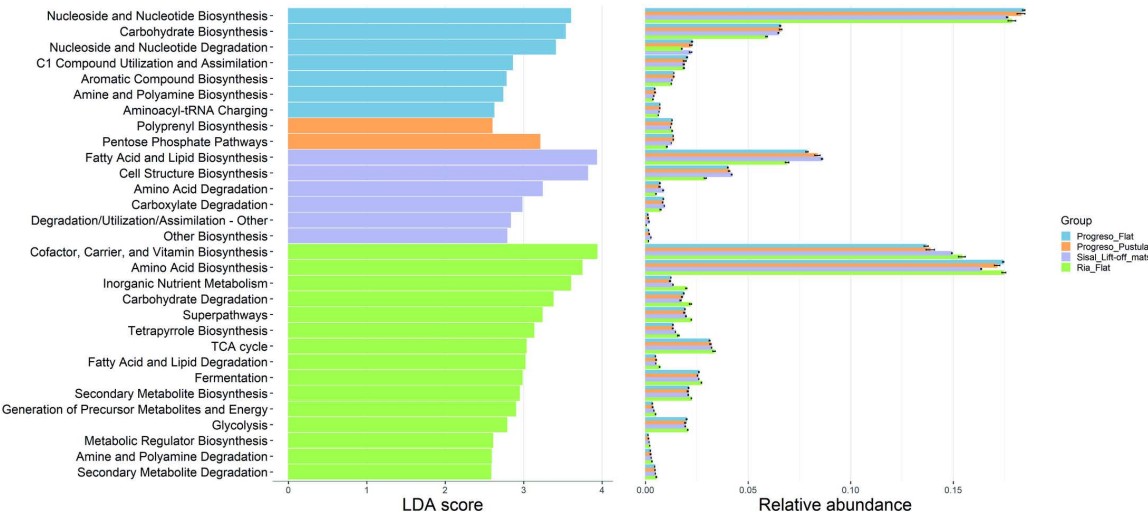

**Fig 6. Predicted pathways of microbial communities.** LefSe analysis performed on the functional prediction of microbial communities from microbial mat samples using PICRUSt2.

## Discussion

This study presents the first characterization of microbial mats from the coastal karstic habitats in the YP, revealing key microbial assemblages and their ecological significance. By integrating high-throughput sequencing data with physico-chemical analyses, we provide novel insights into the microbial diversity of these mats and their relationship with environmental drivers.

### Physicochemical characteristics of mats and prokaryotic community structure

Previous studies on microbial mats in the YP have primarily focused on microbialites in Laguna Bacalar [34], freshwater cenotes [35], and the Chicxulub crater [36], which differ markedly from the coastal mats analyzed in this study. For instance, microbial mats in cenotes have been found to contain Nitrospirae and Acidobacteria, groups that were practically absent in our study, likely due to differences in salinity and nutrient availability [35]. The high salinity of our sites, particularly in Ría Lagartos, suggests similarities with hypersaline mats such as those found in Shark Bay, Australia. However, our results indicate that while some taxonomic groups are shared, such as Cyanobacteria, Chloroflexi, Proteobacteria and Bacteroidetes, the microbial composition in the YP mats is shaped by distinct environmental factors, particularly salinity, redox potential, and proximity to mangrove ecosystems. Additionally, halite and calcite were present in all studied microbial mats. These findings may be explained by the karstic characteristics of the region.

Benthic microbial mats, without much vertical relief, are commonly known as smooth mats [1–6]. This type of macrostructure was observed in Progreso, close to degraded mangroves, and at Ría Lagartos in the absence of mangrove ecosystems. In contrast, pustular mats have a tufted structure, which distinguishes them, albeit somewhat subjectively, from flat mats [8]. Pustular mats were found at Progreso, associated with mangrove ecosystems with hydrological rehabilitation, consisting of constructing canals to facilitate water recirculation [19]. It has been suggested that as mangroves recover, there is a progression from smooth to pustular mats, which can be visualized as columnar growth [4,10]. Additionally, mat formations are influenced by various environmental factors, such as sediment influx, substrate elevation, interstitial groundwater, waves or currents, and hydroperiod, among others [5,7]. Mats can even become separated from the benthic substrate. Physical disruption caused by temperature, gas production, or flooding can release the mat from the

sediment, resulting in a floating mass in water. Lift-off mats have been discovered in salt lakes [37], caves, and hot springs [38]. In this study, we report the presence of lift-off mats occurring at low salinity next to degraded mangroves in Sisal.

Salinity, temperature, and redox potential were the environmental variables that explained the variance in the taxonomic composition of the microbial mat communities. It is well known that those variables are key drivers of microbial populations in many environments. For example, salinity triggers osmotic stress, requiring a complex metabolic specialization [39]. The negative correlation between salinity and alpha diversity was evident, with the highest salinity observed in Ría Lagartos resulting in significantly lower diversity. This pattern is consistent with previous studies in terrestrial and marine ecosystems. Temperature, through its overarching role in regulating overall microbial activity and biogeochemical cycling rates, also strongly influences microbial community composition [40]. Finally, redox potential controls and is controlled by available electron donors and acceptors, and the presence or absence of microbes capable of using these electron donors and acceptors for metabolism. [41].

## The potential role of microbial mats in mangrove ecosystems

Mangrove ecosystems usually develop in the intertidal zone of tropical coastlines, where benthic microbial mats may also be found. Our results highlight clear differences in microbial community composition between mats associated with degraded versus rehabilitated mangroves. In Progreso, rehabilitated mangrove-associated mats were dominated by Chloroflexi and Planctomycetes, both known for their roles in organic matter degradation and nitrogen cycling. In contrast, mats in degraded mangrove zones, such as Sisal, exhibited a higher abundance of Cyanobacteria and Gammaproteobacteria, taxa often linked to primary production, nitrogen and sulfur cycling. This suggests that microbial mats in degraded mangroves may function as early-stage bioengineers, facilitating ecosystem recovery by contributing to nutrient cycling and organic matter stabilization. These observations were also supported by the metabolic prediction of microbial communities, where amino acid metabolism, nucleotides and carbohydrate biosynthesis were observed in microbial mats from Sisal and Progreso associated with degraded mangrove zones. Microbial mats in mangrove ecosystems influence nitrogen and carbon cycling. They enhance nitrogen fixation in sediment [42], integrate with mangrove roots [43], either provide or compete for nitrogen depending on habitat type [44], and produce exopolysaccharides [45]. Recent studies using 16S rRNA gene sequencing in mangrove associated mats have demonstrated that microbial community shifts can be used as indicators of ecosystem health, with particular emphasis on sulfur-oxidizing, nitrogen-fixing bacteria and mineral forming bacteria [46–48]. Undoubtedly, the contribution of microbial mats to the biogeochemical cycling of mangrove ecosystems remains understudied. The participation of benthic microorganisms in the biogeochemical cycling of mangrove habitats should be considered in conservation and rehabilitation/recovery efforts in mangrove ecosystems [49].

## Prokaryotic diversity of flat, pustular, and lift-off mats

The microbial community structure of both flat and pustular mats from Progreso was relatively similar across all samples (Figs 3 and 4). These communities were mainly dominated by Bacteroidia, Chloroflexi, Planctomycetes, and Proteobacteria, which is consistent with observations from other microbial mats across the world, including those from Guerrero Negro [50], Elkhorn Slough [51] and Shark Bay [10]. A key finding of this study is the recurrence of dominant microbial groups across different microbial mat types. This pattern has been observed in well-characterized microbial mats such as those in Guerrero Negro and Elkhorn Slough. However, taxonomic similarity does not necessarily equate to functional redundancy. Within each mat, different taxa may carry metabolic pathways, leading to differential functional capacities even among similar microbial assemblages [52]. Our ASV data indicates that significant differences exist between mat types at the family and genus levels. For example, Ría Lagartos mats, while dominated by Halobacteria at the class level, show unique taxonomic profiles at the genus level, with distinct contributions from *Salinibacter* and Balneolaceae, suggesting a distinct adaptation to hypersaline conditions. Additionally, Proteobacteria (Ectothiorhodospiraceae, Chromatiaceae) in Ría

Lagartos indicate an active sulfur cycle, distinguishing it from Progreso and Sisal. In contrast, Progreso exhibits a higher prevalence of Chloroflexi and Bacteroidota, likely contributing to organic matter degradation.

Interestingly, flat mats from Ría Lagartos were dominated by the archaeal clade Halobacteria, Cyanobacteria, and Rhodothermia (Fig 4). These microbial communities are more similar to those reported in endoevaporites and salt crusts than those reported from microbial mats [53], which is consistent with the high salinity conditions of the site (close to 10%). Halobacteriales constitute a clade of microorganisms found in hypersaline environments worldwide with key metabolic capabilities, including photoautotrophy, carbon fixation, aerobic methane oxidation and denitrification [54]. Cyanobacteria is a clade that includes primary producers [3]. Additionally, lift-off mats from Sisal were dominated by Bacteroidia, Gammaproteobacteria, and Cyanobacteria (Fig 4). Information on lift-off mats from coastal environments is limited. Jhon and Paton [37] investigated lift-off mats at hypersaline conditions in coastal lakes of Western Australia, reporting different cyanobacteria, such as *Aphanothece*, *Oscillatoria*, and *Microcoleus*, using culture-dependent techniques. Our work provides important information on the dominant microbial taxa occurring in coastal lift-off mats, but temporal studies are required to understand the ecological succession of lift-off mats, flat, and pustular mats on these sites.

The organisms with higher LDA values resulted from the LefSe analysis (employed to identify the most representative ASVs present in each of the microbial mats studied) indicated that Candidatus Chlorothrix, uncultured Phycisphaeraceae and AKAU3564 characterized the flat mats from Progreso. *Chlorothrix*, a, sulfide-dependent anoxygenic photoautotrophic bacterium, has also been observed in similar hypersaline microbial mats from Guerrero Negro [50,55]. Phycisphaeraceae, comprising facultative anaerobes, are present in sediments that perform anaerobic ammonium oxidation dependent on natural organic matter [56]. The AKAU3564 clade is frequently found in marine sediments and is proposed to be associated with methane-oxidizing microorganisms [57]. The pustular mats from Progreso displayed SBR1031, uncultured *Phaeodactylibacter*, and uncultured Ardenticatenales. The environmental group SBR1031, also known as Aggregatilineales, belongs to the Chloroflexota phylum, commonly found in extreme environments [58]. Phaeodactylibacter has been reported as aerobic heterotrophs and fermenters in marine environments [59]. Members of Ardenticatenales have also been observed in coastal hot springs, being capable of using nitrate or ferric iron as electron acceptors [60]. Salinibacter, Balneolaceae, and Coleofasciculus_PCC_7420 dominated the flat mats from Ría Lagartos. *Salinibacter* is a fermentative extreme halophile bacterium repeatedly found in hypersaline ecosystems, particularly reported in solar salterns [61]. *Balneolaceae* is a family frequently identified in salt mines, solar salterns, or marine ecosystems, with heterotrophic metabolism involved in carbon degradation. *Coleofasciculus* (formerly classified as *Microcoleus*) is a global cyanobacterium known as a mat builder due to its production of polymeric substances [3]. Finally, the lift-off mats from Sisal were represented by *Thioalkalovibrio*, *Robiginitalea*, and uncultured Lentimicrobiaceae. *Thioalkalovibrio* is a chemolithoautotrophic clade that reduces sulfur compounds [62]. *Robiginitalea* is a genus of bacteria known to be chemoheterotrophs (obtain their energy from organic matter), and they have the ability to metabolize complex, high molecular weight compounds [63]. Members of Lentimicrobiaceae are considered to be strictly anaerobic fermenters that are not capable of breaking down complex carbon compounds [64].

## Conclusions

This work represents the first study to identify and quantify the prevalence of coastal microbial mats in the Yucatan Peninsula. Our findings shed new light on the distribution and ecological importance of these unique microbial communities, which may play a crucial role in shaping the region's coastal ecosystems since some of them were associated with mangrove forest ecosystems. Our findings showed a diverse array of microbial macrostructures, which have been previously classified as flat, lift-off mats, and pustular microbial mats. Each of these distinct types of mats exhibited unique physical and chemical properties and significantly different microbial communities. Our analysis also revealed that the community structure of these mats is strongly influenced by salinity, redox potential, and temperature, indicating the significance of physicochemical micro-gradients in shaping microbial diversity. By characterizing these mats'

distribution, environmental characteristics, and microbial biodiversity, this work expands our understanding of microbial ecosystems in coastal habitats and highlights the importance of these complex systems for the health and functioning of coastal karst ecosystems. Future studies should focus on metagenomic and transcriptomic analyses to better understand functional pathways within these microbial communities. Additionally, long-term monitoring of microbial mats in rehabilitated mangrove areas could help assess their role in ecosystem restoration. Given their importance in biogeochemical cycling, microbial mats should be considered in conservation and management strategies for coastal environments in the Yucatan Peninsula.

## Supporting information

**S1 Table. Mineral composition of microbial mats.** Comparative table of the mineral composition of studied microbial mats from the Yucatan Peninsula. (x) presence, (-) absence.
(DOCX)

**S2 Table. 16s rRNA gene sequencing stats.** Sequencing stats before and after denoising and chimera remotion.
(XLSX)

**S3 Table. PERMANOVA between environmental variables and microbial diversity.** Statistical analysis on the UniFrac distance matrix, using 16S rRNA gene sequences from microbial mats and corresponding environmental data. Asterisks denote statistically significant data.
(DOCX)

**S4 Table. Observed microbial diversity.** Classified ASV's from 16s rRNA gene sequencing of microbial mats.
(XLSX)

**S5 Table. PICRUSt2 table.** Functional analysis of microbial communities using PICRUSt2 predictions.
(TSV)

**S1 Fig. Metabolic activity across microbial mat type.** Comparative analysis of metabolic pathways across microbial mat types, highlighting amino acid, methane, sulfur, and carbon fixation metabolisms.
(TIF)

## Acknowledgments

We are grateful to the technicians from the Primary Productivity Laboratory from CINVESTAV-Mérida for their help during the fieldwork. Thanks to Tania Cota and Anabel Suárez-Guevara for the contribution in the map and figures. Santiago Cadena thanks to CONACHYT for the postdoctoral fellowship 570049 (2024–2025).

## Author contributions

**Conceptualization:** Santiago Cadena, Claudia Teutli-Hernández, Jorge A. Herrera-Silveira, M. Leopoldina Aguirre-Macedo, Luisa I. Falcón, Brad M. Bebout, José Q. García-Maldonado.

**Data curation:** Santiago Cadena.

**Formal analysis:** Santiago Cadena.

**Funding acquisition:** Claudia Teutli-Hernández, José Q. García-Maldonado.

**Investigation:** Santiago Cadena, Claudia Teutli-Hernández, Jorge A. Herrera-Silveira, M. Leopoldina Aguirre-Macedo, Luisa I. Falcón, Brad M. Bebout, José Q. García-Maldonado.

**Methodology:** Santiago Cadena, Claudia Teutli-Hernández.

**Project administration:** Claudia Teutli-Hernández, José Q. García-Maldonado.

**Resources:** Claudia Teutli-Hernández, José Q. García-Maldonado.

**Software:** Santiago Cadena, José Q. García-Maldonado.

**Supervision:** Claudia Teutli-Hernández, Jorge A. Herrera-Silveira, M. Leopoldina Aguirre-Macedo, Luisa I. Falcón, Brad M. Bebout, José Q. García-Maldonado.

**Validation:** Santiago Cadena, José Q. García-Maldonado.

**Visualization:** Santiago Cadena.

**Writing – original draft:** Santiago Cadena.

**Writing – review & editing:** Santiago Cadena, Claudia Teutli-Hernández, Jorge A. Herrera-Silveira, M. Leopoldina Aguirre-Macedo, Luisa I. Falcón, Brad M. Bebout, José Q. García-Maldonado.

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
