## [Decision Letter · Decision Letter 0]

PONE-D-24-48408Microbial diversity of coastal microbial mats formations in karstic habitats from the Yucatan Peninsula, MexicoPLOS ONE

Dear Dr. García-Maldonado,

Thank you for submitting your manuscript to PLOS ONE. After careful consideration, we feel that it has merit but does not fully meet PLOS ONE’s publication criteria as it currently stands. Therefore, we invite you to submit a revised version of the manuscript that addresses the points raised during the review process.

We look forward to receiving your revised manuscript.

Kind regards,

Satheesh Sathianeson, Ph.D

Academic Editor

PLOS ONE

Journal Requirements:

“This study was funded by Consejo Nacional de Ciencias, Humanidades y Tecnologías (CONAHCYT) project CF-2019-848287 awarded to JQGM. 

https://conahcyt.mx/.

In addition, sampling was performed thanks to Programa de Apoyo a Proyectos de Investigación e Innovación Tecnológica (PAPIIT-UNAM) IN216219 awarded to CTH.

https://dgapa.unam.mx/”

4. We note that Figure 1 in your submission contain map/satellite image which may be copyrighted. All PLOS content is published under the Creative Commons Attribution License (CC BY 4.0), which means that the manuscript, images, and Supporting Information files will be freely available online, and any third party is permitted to access, download, copy, distribute, and use these materials in any way, even commercially, with proper attribution. For these reasons, we cannot publish previously copyrighted maps or satellite images created using proprietary data, such as Google software (Google Maps, Street View, and Earth). For more information, see our copyright guidelines: http://journals.plos.org/plosone/s/licenses-and-copyright.

5. We note that Figure 1 in your submission contain copyrighted images. All PLOS content is published under the Creative Commons Attribution License (CC BY 4.0), which means that the manuscript, images, and Supporting Information files will be freely available online, and any third party is permitted to access, download, copy, distribute, and use these materials in any way, even commercially, with proper attribution. For more information, see our copyright guidelines: http://journals.plos.org/plosone/s/licenses-and-copyright.

Reviewers' comments:

Reviewer's Responses to Questions

**Comments to the Author**

1. Is the manuscript technically sound, and do the data support the conclusions?

Reviewer #1: Yes

Reviewer #2: Yes

2. Has the statistical analysis been performed appropriately and rigorously? 

Reviewer #1: Yes

Reviewer #2: Yes

3. Have the authors made all data underlying the findings in their manuscript fully available?

Reviewer #1: Yes

Reviewer #2: Yes

4. Is the manuscript presented in an intelligible fashion and written in standard English?

Reviewer #1: Yes

Reviewer #2: Yes

5. Review Comments to the Author

Reviewer #1: The work "Microbial diversity of coastal microbial mats formations in karstic habitats from the Yucatan Peninsula, Mexico" describes bacterial diversity of microbial mat communities. It is technically sound, and the applied statistical methods are correct. Raw sequence data is available. But several issues should be corrected.

Regarding the introduction, it is very brief and fails to convey the importance of the study. It defines microbial mats, types, and karstic habitats citing old references. It is stated (L 67): "...karstic microbial mats have received comparatively less attention than other ecosystems". How are your karst coastal mats different from other coastal ecosystems? You even mention the formation of salterns, so perhaps your mats are somehow related to hypersaline mats? Any reference to that? A brief google search "karst microbial mats" directs me to caves and springs, but this is not the karst habitat you describe. Perhaps a better description of your sites in addition to general references would be helpful.

The discussion has similar problems, it is shallow. A few suggestions to improve it:

You cite other studies on YP. How are they similar/different to your own?

Mangroves are mentioned in a whole paragraph, but mostly citing old (more than 10 years) references. I'm sure there are several more recent works related to mangroves and mats, or at least, related microbial communities described by sequencing, which could add to your discussion.

Going further, given that many microbial mats and communities have been described using sequencing, I'd ask "what does my data look like?" You do mention well-described systems such as Shark Bay and a few others. Anything else? Any others with the same amplicon you used that you could compare to?

How about functions? Are your taxonomic differences conserved functionally? You might use tools such as picrust or tax4fun.

Other minor issues are below:

Line 58: reference 10 does not analyze archaeal communities. Please correct text or reference.

Line 111: reference 17 does not belong here. From here on, references seem to be "moved": 17 belongs to 18 on the ref list, 18 belongs to 19, etc.

Line 182: The data in the table is mentioned in the text, I think this table is not necessary. You might include it as Supplementary.

Line 235-236: "Pustular mats were found at Progreso, associated with mangrove ecosystems with hydrological rehabilitation, that consist in the construction of canals to facilitate water input" Why is ref 8 cited here?

Reference 34 is not cited

Line 283: The fact that several well-studied mat systems present similar taxonomic groups is not an argument against functional redundancy, in my humble opinion.

First, most likely you have redundancy within each system, translating into differential abundances of groups performing similar functions on each site. Second, you can find similar functions in quite different systems performed by different organisms (In addition to ref 45, see https://peerj.com/articles/17412/). I suggest you discuss this further or change this sentence.

Reviewer #2: Cadena and collaborators describe the microbial composition of microbial mat formations in karstic habitats and the differences in their diversity.

The manuscript is well written and uses statistics to relate basic environmental variables to the observed differences amongst the explored habitats. The manuscript mentions differences of mangroves health status (degraded and restored), however, they fail to mention what is considered a degraded and a restored mangrove. Also, their discussion on the composition of the microbial communities skips the relationship of these communities with the health status of the mangroves and it separates the effects of the physical-chemical aspects of the environments into a different discussion. I recommend improving the discussion by including all aspects that can be attributed to the changes observed in the community altogether. I.e. include the location, physical and chemical characteristics, and degraded or restored status together when discussing the differences between communities.

Also, I suggest including a wider paragraph on the different functions that could be attributed to these communities since the presented discussion only suggest broad participation in some geochemical cycles but no specific functionalities within them.

I have included more specific comments in the pdf of the manuscript file attached.

In all, the manuscript presents important information of interest to the microbial ecology community. However, a deeper insight into the differences between the habitats and the environmental variables causing it, together with the health of the ecosystem they were found in would be recommended. Considering the differences between these mats and the mangrove sediments they belong to would also improve the quality of the discussion.

6. PLOS authors have the option to publish the peer review history of their article (what does this mean?). If published, this will include your full peer review and any attached files.

Reviewer #1: **Yes:**

Reviewer #2: No

---

## [Author Response · Author response to Decision Letter 1]

3 Mar 2025

Reviewer #1: The work "Microbial diversity of coastal microbial mats formations in karstic habitats from the Yucatan Peninsula, Mexico" describes bacterial diversity of microbial mat communities. It is technically sound, and the applied statistical methods are correct. Raw sequence data is available. But several issues should be corrected.

Answer: We appreciate the positive comments made on our manuscript and thank you for your careful review and constructive suggestions.

Regarding the introduction, it is very brief and fails to convey the importance of the study. It defines microbial mats, types, and karstic habitats citing old references. It is stated (L 67): "...karstic microbial mats have received comparatively less attention than other ecosystems". How are your karst coastal mats different from other coastal ecosystems? You even mention the formation of salterns, so perhaps your mats are somehow related to hypersaline mats? Any reference to that? A brief google search "karst microbial mats" directs me to caves and springs, but this is not the karst habitat you describe. Perhaps a better description of your sites in addition to general references would be helpful.

Answer: We thank reviewer for these observations. In this version of the manuscript, we have modified the introduction (lines 67-93). We have expanded the information on the background of microbial mats in karst ecosystems and have provided a better description of the karst environments we are investigating. We have also highlighted the relevance of this study.

“While some studies have investigated microbial diversity and function in hypersaline ecosystems, microbial mats in karstic environments have received comparatively less attention. Kolda and colleagues [13] investigated microbial mats from the Krčić spring, Croatia, finding that Cyanobacteria constituted the core structure of the community, reporting Microcoleus and Phormidium as the most abundant cyanobacterial taxa, followed by several groups of Alphaproteobacteria and Firmicutes. Mulec and Engel [14] studied the microbial diversity of mats along an oxygen-sulphide gradient in the Žveplenica karst spring in Slovenia, finding abundant sulphur-metabolizing taxa such as Thiothrix, Azospira, Iodobacter and Thiobacillus. Microbial mats from karst springs in Lower Kane Cave, Wyoming, were investigated using 16S rRNA gene clone libraries, reporting chemolithoautotrophic taxa related to Epsilon, Gamma and Beta Proteobacteria, as well as Acidobacterium and Chlorobi [15]. Compared to karstic microbial mats studied in caves and springs, coastal mats are exposed to different stressors, such as tidal fluctuations, salt intrusion, and mangrove-derived organic matter [1-3], which provide a distinct ecological niche for microbial mats.

The Yucatan Peninsula (YP) is one of the most extensive karst systems known on the planet, encompassing a limestone platform of 165,000 km2, with a network of cenotes, caves, and coastal lagoons [16]. The climatic conditions of the YP are divided into three seasons: dry (March-June), rainy (July-October), and north-winds (November-February). Coastal karstic habitats in the YP are influenced by dynamic environmental factors, such as salinity gradients, seasonal precipitation, and the proximity of mangrove ecosystems [17]. The hypersaline conditions resulting from high evaporation rates in these coastal zones often lead to the formation of solar salterns, which may harbor microbial mats similar to those in other hypersaline environments [18]. However, the specific microbial communities inhabiting these karstic mats remain poorly characterized. This study explores the prokaryotic diversity of microbial mats in three coastal karstic habitats in the Yucatan Peninsula: Sisal, Progreso, and Ría Lagartos. Using high-throughput sequencing of the 16S rRNA gene, we characterize the microbial community composition and identify key environmental drivers, including salinity, temperature, and redox potential. By comparing these karstic mats with other coastal and hypersaline ecosystems, this research aims to provide new insights into microbial mats' ecological and functional roles in understudied karstic coastal habitats, contributing to broader efforts to conserve and manage these unique ecosystems.”

The discussion has similar problems, it is shallow. A few suggestions to improve it:

You cite other studies on YP. How are they similar/different to your own?

Mangroves are mentioned in a whole paragraph, but mostly citing old (more than 10 years) references. I'm sure there are several more recent works related to mangroves and mats, or at least, related microbial communities described by sequencing, which could add to your discussion. Going further, given that many microbial mats and communities have been described using sequencing, I'd ask "what does my data look like?" You do mention well-described systems such as Shark Bay and a few others. Anything else? Any others with the same amplicon you used that you could compare to?

Answer: We sincerely appreciate the reviewer's insightful comments regarding the depth of the discussion. In implementing these revisions, we have rewritten the entire discussion section. We believe the discussion now provides a more robust and comprehensive interpretation of our results, making clearer comparisons to existing literature and emphasizing the novelty and ecological significance of our findings.

How about functions? Are your taxonomic differences conserved functionally? You might use tools such as picrust or tax4fun.

Answer: We thank reviewer for this observation. In order to better understand metabolic functionality of microbial mats, we have performed PICRUSt2 analysis to our samples. This information can be found in the following sections.

Methods, lines 158-164:

The PICRUSt2 v.2.2.0 [33] pipeline was used to infer metagenomic functional content based on QIIME2 microbial community profiles obtained from representative sequences and sequence features. Predicted functional genes were categorized using the KEGG ontology database (http://www.genome.jp/kegg/). STAMP software (v2.1.3) was used for statistical analyses comparing community structures between localities at different levels (Welch's t-test); and LEfSe analysis was used to assess differential functional abundance based on predicted KEGG pathways (LDA score 4.0 and p-value < 0.05).

Results, lines 238-248:

Functional prediction of microbial communities derived from PICRUSt2 analysis allowed to identify 409 metabolic pathways among all microbial mat samples (Supplementary Table S5). Figure 6 shows the predicted metabolic pathways for each microbial mat type, revealed by LefSe analysis. Flat mats from Progreso showed higher proportions of nucleosides, nucleotides and carbohydrates biosynthesis, while Pustular mats from Progreso showed polyprenyl biosynthesis and pentose phosphate pathways. Lift-off mats from Sisal possessed differential abundances of fatty acids, lipids and cell structure biosynthesis. Flat mats from Ría Lagartos were characterized by vitamins, cofactors and, amino acid biosynthesis, inorganic nutrient metabolism and carbohydrate degradation. In addition, amino acid metabolism was significantly higher in lift-off mats from Sisal. Methane metabolism was statistically significant in flat mats from Ría Lagartos, and sulfur and carbon fixation metabolisms were not statistically different among microbial mat types (Supplementary Figure S6).

Discussion, lines 303-306.

These observations were also supported by the metabolic prediction of microbial communities, where amino acid metabolism, nucleotides and carbohydrate biosynthesis were observed in microbial mats from Sisal and Progreso associated with degraded mangrove zones. Microbial mats in mangrove ecosystems influence nitrogen and carbon cycling.

Other minor issues are below:

Line 58: reference 10 does not analyze archaeal communities. Please correct text or reference.

Answer: We thank this observation. We have added the correct reference.

Line 111: reference 17 does not belong here. From here on, references seem to be "moved": 17 belongs to 18 on the ref list, 18 belongs to 19, etc.

Answer: We appreciate this observation. In this version of the manuscript we have ensured that the references are in accordance with the text.

Line 182: The data in the table is mentioned in the text, I think this table is not necessary. You might include it as Supplementary.

Answer: We have eliminated this table from the main manuscript and added as supplementary material.

Line 235-236: "Pustular mats were found at Progreso, associated with mangrove ecosystems with hydrological rehabilitation, that consist in the construction of canals to facilitate water input" Why is ref 8 cited here?

Answer: We appreciate this observation. In this version of the manuscript we have ensured that the references are in accordance with the text.

Reference 34 is not cited

Answer: We thank this observation. Reference 34 was eliminated.

Line 283: The fact that several well-studied mat systems present similar taxonomic groups is not an argument against functional redundancy, in my humble opinion.

First, most likely you have redundancy within each system, translating into differential abundances of groups performing similar functions on each site. Second, you can find similar functions in quite different systems performed by different organisms (In addition to ref 45, see https://peerj.com/articles/17412/). I suggest you discuss this further or change this sentence.

Answer: We appreciate these insightful comments. This section of the manuscript was rewritten in lines 322-330.

“However, taxonomic similarity does not necessarily equate to functional redundancy. Within each mat, different taxa may carry metabolic pathways, leading to differential functional capacities even among similar microbial assemblages [53]. Our ASV data indicates that significant differences exist between mat types at the family and genus levels. For example, Ría Lagartos mats, while dominated by Halobacteria at the class level, show unique taxonomic profiles at the genus level, with distinct contributions from Salinibacter and Balneolaceae, suggesting a distinct adaptation to hypersaline conditions. Additionally, Proteobacteria (Ectothiorhodospiraceae, Chromatiaceae) in Ría Lagartos indicate an active sulfur cycle, distinguishing it from Progreso and Sisal. In contrast, Progreso exhibits a higher prevalence of Chloroflexi and Bacteroidota, likely contributing to organic matter degradation.”

Reviewer #2: Cadena and collaborators describe the microbial composition of microbial mat formations in karstic habitats and the differences in their diversity.

The manuscript is well written and uses statistics to relate basic environmental variables to the observed differences amongst the explored habitats. The manuscript mentions differences of mangroves health status (degraded and restored), however, they fail to mention what is considered a degraded and a restored mangrove. Also, their discussion on the composition of the microbial communities skips the relationship of these communities with the health status of the mangroves and it separates the effects of the physical-chemical aspects of the environments into a different discussion. I recommend improving the discussion by including all aspects that can be attributed to the changes observed in the community altogether. I.e. include the location, physical and chemical characteristics, and degraded or restored status together when discussing the differences between communities.

Also, I suggest including a wider paragraph on the different functions that could be attributed to these communities since the presented discussion only suggest broad participation in some geochemical cycles but no specific functionalities within them.

We thank the reviewer for all valuable and very helpful comments while revising our paper. We have studied comments carefully and have made corrections to address these issues. We have now explicitly defined what is considered a degraded and restored mangrove within our study sites. These definitions have been incorporated into both the Methods and Discussion sections to ensure clarity. We have revised the Discussion to integrate microbial community composition with both physicochemical variables and mangrove health status. We appreciate the reviewer’s insightful comments, which have significantly strengthened our manuscript.

We have included more specific comments in the pdf of the manuscript file attached.

Lines 83-84. How is degraded defined? How have they been restored? What defines a restored mangrove?

Answer: We have defined degrades and restored mangroves in lines 99-103:

“Degraded mangroves refer to areas that previously supported mangrove vegetation but had widespread loss, resulting in the absence of mangrove cover. Restored mangroves are sites where previously degraded areas have undergone ecological rehabilitation, characterized by the reestablishment of mangrove seedlings through active restoration efforts”.

Line 92. How can this water be analyzed without exposing it to atmospheric air? Mainly for redox potential measurements which can change rapidly when in contact with O2.

Answer: We agree with this comment. Interstitial water was collected using a 50 ml syringe coupled to an acrylic tube sealed at the bottom but containing a small hole. This tube-syringe system was inserted into mats and water was extracted by suction and measured immediately in situ. We have detailed this information in methodology (lines 106-111).

“At each sampling site, interstitial water was extracted using a specialized device consisting of a 50 ml syringe connected to an acrylic tube. The tube, sealed at the bottom except for a small hole (3 mm diameter), was inserted to a depth of 30 cm within the mats. Water was drawn out by syringe suction, yielding 40 ml per sample while minimizing exposure to air [20]. Immediately following collection, physicochemical properties, including salinity, temperature, pH, and redox potential, were recorded using a portable multi-parameter instrument”

124. Utilized

Answer. It was corrected.

Line 126. Why Silva

Answer: Silva database was selected because this database is one of the most updated and extensive collection of 16S sequences. This information was incorporated as follows (Lines 146-148):

“Representative ASVs were assigned using the SILVA small subunit ribosomal RNAs (16S) v138 database as a reference, since SILVA provides one of the most updated and extensive collections of 16S gene sequences derived from publicly available datasets [25], through the V-SEARCH plugin [26]”.

139. Same as above, how do you define these two?

Answer: The complete description of these types of mats are explained in lines 270-282:

“Benthic microbial mats without much vertical relief, are commonly known as smooth mats [1-6]. This type of macrostructure was observed in Progreso, close to degraded mangroves, and at Ría Lagartos in the absence of mangrove ecosystems. In contrast, pustular mats have a tufted structure, which distinguishes them, albeit somewhat subjectively, from flat mats [8]. Pustular mats were found at Progreso, associated with mangrove ecosystems with hydrological rehabilitation, that consist in the construction of canals to facilitate water input [7-8]. It has been suggested that as mangroves recover, there is a progression from smooth to pustular mats, which can be visualized as columnar growth [4,10]. Additionally, mat formations are influenced by a variety of environmental factors, such as sediment influx, substrate elevation, interstitial groundwater, waves or currents, hydroperiod, among others [5, 7]. Mats can even become separated from the benthic substrate. Physical disruption caused by temperature, gas production, or flooding can release the mat from the sediment, resulting in a floating mass in water. These so-called lift-off mats have been discovered in salt lakes [31], caves [32] and hot springs [33]. In this study, we report the presence of lift-off mats occurring at low salinity next to degraded mangroves in Sisal. “

Line 191. By diverse you mean an index? Because if you are referr

---

## [Decision Letter · Decision Letter 1]

PONE-D-24-48408R1Microbial diversity of coastal microbial mats formations in karstic habitats from the Yucatan Peninsula, MexicoPLOS ONE

Dear Dr. García-Maldonado,

Thank you for submitting your manuscript to PLOS ONE. After careful consideration, we feel that it has merit but does not fully meet PLOS ONE’s publication criteria as it currently stands. Therefore, we invite you to submit a revised version of the manuscript that addresses the points raised during the review process.

We look forward to receiving your revised manuscript.

Kind regards,

Satheesh Sathianeson, Ph.D

Academic Editor

PLOS ONE

Journal Requirements:

Reviewers' comments:

Reviewer's Responses to Questions

**Comments to the Author**

1. If the authors have adequately addressed your comments raised in a previous round of review and you feel that this manuscript is now acceptable for publication, you may indicate that here to bypass the “Comments to the Author” section, enter your conflict of interest statement in the “Confidential to Editor” section, and submit your "Accept" recommendation.

Reviewer #1: (No Response)

Reviewer #2: All comments have been addressed

2. Is the manuscript technically sound, and do the data support the conclusions?

Reviewer #1: Yes

Reviewer #2: Yes

3. Has the statistical analysis been performed appropriately and rigorously? 

Reviewer #1: Yes

Reviewer #2: Yes

4. Have the authors made all data underlying the findings in their manuscript fully available?

Reviewer #1: Yes

Reviewer #2: Yes

5. Is the manuscript presented in an intelligible fashion and written in standard English?

Reviewer #1: Yes

Reviewer #2: Yes

6. Review Comments to the Author

Reviewer #1: The revised version from the work "Microbial diversity of coastal microbial mats formations in karstic habitats from the Yucatan Peninsula, Mexico" has significantly improved.

All the problems were adressed and the manuscript looks good.

I only found a newly added sentence that might be incorrect

L343-344: "Halobacteriales constitute a clade found in hypersaline environments worldwide with autotrophic capabilities". I think this is not right, or if it is, please provide a reference. As far as I know, autotrophy is not demonstrated in this clade.

Reviewer #2: All questions and concerns from both reviewers have been addressed. The discussion section has now been rewritten and provides a better understanding of the differences between studied microbial communities and their causes.

7. PLOS authors have the option to publish the peer review history of their article (what does this mean?). If published, this will include your full peer review and any attached files.

Reviewer #1: **Yes: **

Reviewer #2: No

---

## [Author Response · Author response to Decision Letter 2]

30 Apr 2025

We are grateful for the opportunity to submit our revised article titled: “Microbial diversity of coastal microbial mats formations in karstic habitats from the Yucatan Peninsula, Mexico” (PONE-D-24-48408). We truly appreciate the positive remarks and helpful feedback provided by the reviewer. We have addressed his concerns and added reference as suggested. In addition, we have reviewed the cited literature of the article, ensuring the references are complete and correct, and switching the retracted article by the corrected version in reference number 32.

Reviewer 1 comments:

All the problems were adressed and the manuscript looks good.

I only found a newly added sentence that might be incorrect

L343-344: "Halobacteriales constitute a clade found in hypersaline environments worldwide with autotrophic capabilities". I think this is not right, or if it is, please provide a reference. As far as I know, autotrophy is not demonstrated in this clade.

Answer: We appreciate this comment from the reviewer. We agree that “autotrophic” may cause confusion, then, we have detailed the metabolic functions of Halobacterales in hypersaline ecosystems and added the reference.

Halobacteriales constitute a clade of microorganisms found in hypersaline environments worldwide with key metabolic capabilities, including photoautotrophy, carbon fixation, aerobic methane oxidation and denitrification [54].

54. Hazzouri KM, Sudalaimuthuasari N, Saeed EE, Kundu B, et al. Salt flat microbial diversity and dynamics across salinity gradient. Scientific reports. 2022; 12(1):11293. https://doi.org/10.1038/s41598-022-15347-8

---

## [Editor Report · Decision Letter 2]

Microbial diversity of coastal microbial mats formations in karstic habitats from the Yucatan Peninsula, Mexico

PONE-D-24-48408R2

Dear Dr. García-Maldonado,

We’re pleased to inform you that your manuscript has been judged scientifically suitable for publication and will be formally accepted for publication once it meets all outstanding technical requirements.

Kind regards,

Satheesh Sathianeson, Ph.D

Academic Editor

PLOS ONE
---

## [Editor Report · Acceptance letter]

PONE-D-24-48408R2

PLOS ONE

Dear Dr. García-Maldonado,

I'm pleased to inform you that your manuscript has been deemed suitable for publication in PLOS ONE. Congratulations! Your manuscript is now being handed over to our production team.

Kind regards,

on behalf of

Dr. Satheesh Sathianeson

Academic Editor

PLOS ONE